# The Impact of Lipid Corona on Rifampicin Intramacrophagic Transport Using Inhaled Solid Lipid Nanoparticles Surface-Decorated with a Mannosylated Surfactant

**DOI:** 10.3390/pharmaceutics11100508

**Published:** 2019-10-01

**Authors:** Eleonora Maretti, Cecilia Rustichelli, Magdalena Lassinantti Gualtieri, Luca Costantino, Cristina Siligardi, Paola Miselli, Francesca Buttini, Monica Montecchi, Eliana Leo, Eleonora Truzzi, Valentina Iannuccelli

**Affiliations:** 1Department of Life Sciences, University of Modena and Reggio Emilia, 41125 Modena, Italy; eleonora.maretti@unimore.it (E.M.); cecilia.rustichelli@unimore.it (C.R.); luca.costantino@unimore.it (L.C.); eliana.leo@unimore.it (E.L.); eleonora.truzzi@unimore.it (E.T.); 2Department of Engineering “Enzo Ferrari”, University of Modena and Reggio Emilia, 41125 Modena, Italy; magdalena.gualtieri@unimore.it (M.L.G.); cristina.siligardi@unimore.it (C.S.); paola.miselli@unimore.it (P.M.); monica.montecchi@unimore.it (M.M.); 3Food and Drug Department, University of Parma, 43124 Parma, Italy; francesca.buttini@unipr.it

**Keywords:** solid lipid nanoparticles, mannosylated surfactant, tuberculosis, inhalation, active targeting, pulmonary surfactant

## Abstract

The mimicking of physiological conditions is crucial for the success of accurate in vitro studies. For inhaled nanoparticles, which are designed for being deposited on alveolar epithelium and taken up by macrophages, it is relevant to investigate the interactions with pulmonary surfactant lining alveoli. As a matter of fact, the formation of a lipid corona layer around the nanoparticles could modulate the cell internalization and the fate of the transported drugs. Based on this concept, the present research focused on the interactions between pulmonary surfactant and Solid Lipid Nanoparticle assemblies (SLNas), loaded with rifampicin, an anti-tuberculosis drug. SLNas were functionalized with a synthesized mannosylated surfactant, both alone and in a blend with sodium taurocholate, to achieve an active targeting to mannose receptors present on alveolar macrophages (AM). Physico-chemical properties of the mannosylated SLNas satisfied the requirements relative to suitable respirability, drug payload, and AM active targeting. Our studies have shown that a lipid corona is formed around SLNas in the presence of Curosurf, a commercial substitute of the natural pulmonary surfactant. The lipid corona promoted an additional resistance to the drug diffusion for SLNas functionalized with the mannosylated surfactant and this improved drug retention within SLNas before AM phagocytosis takes place. Moreover, lipid corona formation did not modify the role of nanoparticle mannosylation towards the specific receptors on MH-S cell membrane.

## 1. Introduction

Inhalable powder formulations of drugs intended for immediate biological actions within alveolar macrophages (AM) entail engineering techniques that enable drug emission by Dry Powder Inhaler (DPI) devices, deposition onto alveolar epithelia, and transport into AM [1,2,3]. Actually, most drugs for the treatment of intramacrophagic bacterial infections such as tuberculosis are poorly soluble in water; in this case, their aerosol formulations must be produced by using drugs in the solid state. Untreated drugs exhibit rarely properties suitable for both DPI and targeting to AM, in which *Mycobacterium tuberculosis* resides. Therefore they fail to both reach alveolar epithelium and penetrate AM effectively [4,5]. Among the particle engineering techniques, the recourse to micro- or nanoparticulate drug carriers has the potential to facilitate drug macrophage uptake while maintaining particle respirability and drug dosing accuracy. For these reasons, several potential inhaled powders have been successfully designed using particulate formulation approach. Moreover, particulate formulations are suitable for inserting specific molecules at the particle surface to promote receptor-mediated AM endocytosis [6]. Macrophage mannose receptors (MR), overexpressed on infected AM, can recognize particles functionalized with mannose residues on their surface, promoting their internalization [7,8]. However, the targeting moieties must remain associated on the particle surface also in biological environments without losing their functionality until the target is reached.

The alveoli and consequently AM are lined by the alveolar lining fluid (ALF) that is generated by the alveolar epithelial cells. It is essential for maintaining fundamental lung functions (protection, immune response, and reduction of surface tension at the air-liquid interface) [9,10]. ALF is constituted by the pulmonary surfactant, a surface-active phospholipid/protein mixture mainly constituted by lipids [11,12,13]. Pulmonary surfactant can be adsorbed by particulate matter producing a bio-nano interface referred as protein/lipid corona. Although most of studies focused on the protein/lipid corona formed by interactions of nanoparticles with plasma, more recently some attention has been paid also to other biological fluids such as ALF [13,14,15,16,17,18,19,20,21,22]. Unlike the protein corona formed in plasma, that is attributable to interactions between nanoparticles and proteins, the corona layer formed around inhaled nanoparticles proved to differ markedly, because it preserves a lipid composition regardless of nanoparticle surface properties. This finding was related to the high content of phospholipids (about 80–90%) present in the pulmonary surfactant [12,19]. Such a corona formed around the particles is described to be affected by particle size, shape, surface charge, covalent/coordinate bonding or hydrogen bonding propensity, hydrophobicity, and surface functional groups as well as environment composition and exposure duration. Among these parameters, the surface hydrophobicity and charge density play a more significant role [23,24]. It is known that corona can deeply influence the interaction of nanoparticles with cells and determine not only the nanoparticle fate, but also their properties such as aggregate formation, pulmonary clearance, receptor-mediated transport, and drug release behavior [15,16,18,19,25,26,27], although some discordant results have also been reported. For example, corona layer was found to impair nanoparticle adhesion onto cell surface or to decrease particle uptake by macrophages [16,28,29]. On the other hand, some studies reported an improved cell internalization [14,30,31,32]. Therefore, the implications of corona layer on AM targeting using inhalable carriers remain still an open issue that needs to be further explored.

Within this context, respirable Solid Lipid Nanoparticle assemblies (SLNas) loaded with rifampicin, a clinically useful anti-tuberculosis drug, were previously produced as inhaled pulmonary tuberculosis treatment using DPI device. To this aim, accepted excipients for DPI formulations were processed through an optimized methodology that avoids organic solvents [33,34]. The prototypes were functionalized by means of newly synthesized mannosylated surfactants as AM receptor-specific targeting agents anchored on SLNas surface without the need of chemical reactions [5]. The focus of the present research was the investigation of the interactions between pulmonary surfactant and SLNas functionalized with solely a mannosylated surfactant and also in a blend with the natural bile acid surfactant sodium taurocholate. The effect of such interactions on in vitro drug release and internalization ability by murine alveolar macrophages MH-S cell line was investigated and compared with non-mannosylated SLNas. 

## 2. Materials and Methods

### 2.1. Materials

Palmitic acid (PA) from Fluka Chemie (Buchs, Switzerland), cholesteryl myristate (CM) from TCI Europe (Zwijndrecht, Belgium), polyoxyethylene polyoxypropylene block copolymer (Lutrol F127) (F127) from BASF (Ludwigshafen, Germany), sodium taurocholate (ST) and rifampicin (RIF) from Alfa Aesar (Ward Hill, MA, USA) were purchased. The mannosylated surfactant hexadecanoic acid (aminoethyl α-d-mannopyranoside)amide (MS) was synthesized as previously reported [5] and used as functionalizing/surfactant agent. Simulated Lung Fluid type 3 (SLF) at pH 7.4, used for in vitro drug release, was prepared accordingly to Marques [35]. Curosurf, used for lipid corona determination, was kindly donated by Chiesi Farmaceutici (Parma, Italy). For HPLC analysis, methyl parahydroxybenzoate, used as internal standard (IS), was purchased from Carlo Erba Reagenti (Milan, Italy). For cell culture assays, MH-S cell line was purchased from IZSLER (Brescia, Italy). For cell medium, RPMI 1640, fetal bovine serum (FBS), l-Glutamine, penicillin–streptomycin (P/S), and phosphate-buffered saline (PBS), and trypsin-EDTA from PAN-Biotech (Aidenbach, Germany), d-mannose and dimethyl sulfoxide (DMSO) from VWR International (Milan, Italy) were purchased. All the other chemicals were of analytical grade.

### 2.2. Methods

#### 2.2.1. SLNas Preparation

SLNas were prepared by the melt emulsification technique. In practice, the lipid phase containing 92.5 mg PA, 42.5 mg CM, and 45 mg RIF was melted at 85 °C and emulsified in 10 mL MilliQ water containing the functionalizing/surfactant agent MS (0.1%) alone or in a blend with ST (0.05%) to prepare the samples SLNas/MS or SLNas/MS-ST, respectively. Control samples in which MS was not included, SLNas/ST and SLNas/F127, were obtained by using 0.2% ST and 1% F127, respectively. The two phases were mixed by ultrasonification (Branson Sonifier^®^ SFX150, Emerson US, St. Louis, MO, USA) (150 W for 3 min) and rapidly cooled in ice, under magnetic stirring (15 min). The resulting particle suspension was added with MilliQ water (300 mL) and purified by a dialysis membrane (MWCO 12–14,000 Da; Medicell International Ltd., London, UK) for 30 min to remove the non-encapsulated drug and the excess of functionalizing/surfactant agents. Dialysis time was selected in preliminary studies as the best compromise between drug retaining within SLNas and the surfactant removal. Then, the samples were water diluted 1:55, rapidly frozen at −70 °C in a dry ice/acetone cooling bath, and then freeze-dried (Lio 5P, CinquePascal, Milan, Italy), to obtain a final powder according to the method previously optimized [34]. 

#### 2.2.2. Morphology, Size, and Z-Potential

SLNas morphology was evaluated by Scanning Electron Microscopy (SEM, Nova NanoSEM 450, Fei, Eindhoven, The Netherlands) using TEM mode with STEM detector (30 kV). Carbon/copper TEM grids were soaked in properly diluted SLNas water suspension, dried and coated with carbon under vacuum (carbon coater, Balzers CED-010, Oerlikon Balzers, Balzers, Liechtestein). The obtained images were analyzed by an image software (ImageJ version 1.32J, Java, Bethesda, MD, USA) to measure circularity parameter where the value of a circle is 1, whereas values less than 1 indicate irregular particles. Size, polydispersity index (PDI), and Z-potential values were measured by using Photon Correlation Spectroscopy (PCS) (Zetasizer version 6.12, Malvern Instruments, Worcestershire, UK) equipped with a 4 mW He-Ne laser (633 nm) and a DTS software (Version 5.0). The samples were suspended in MilliQ water for Z-potential analyses and in cell medium for size and PDI determinations. Before the tests, the samples were vortexed for 1 min and sonicated for 3 min in ultrasonic bath (USC200TH, VWR International, Milan, Italy) at 37 °C. Each sample was analyzed three times.

#### 2.2.3. Density, Carr’s Index, and Specific Surface Area

The bulk density of SLNas was determined by measuring the volume of a weighed powder amount in a graduated cylinder. Tapped density was determined by measuring the volume of the tapped powder until no further volume change was observed, according to Ph. Eur. 9.0 [36]. Density data have been used to determine Carr’s Index (compressibility index). True density was determined on the powder using a helium pycnometer (AccuPyc 1330, Micromeritics Inc., Norcross, GA, USA). Specific BET (Brunauer, Emmett and Teller) surface area was determined using the Gemini 2360 instrument from Micromeritics (nitrogen gas). Moreover, a size of SLNas powder, named d(BET), was determined using true density (ρ) and BET specific area as follow: d(BET) = 6/(ρ·Area_BET_). All the measurements were performed in triplicate.

#### 2.2.4. Thermal Analysis

In order to investigate the effect of the preparation process on SLNas physical state, rifampicin, SLNas samples as well as their physical mixture, prepared at the same drug/lipid ratio as SLNas, were analyzed using simultaneous thermogravimetry and differential scanning calorimetry (TG/DSC) (NETZSCH mod. STA409, Netzsch-Gerätebau GmbH, Selb, Germany). A second heating run, in order to simulate SLNas preparation thermal conditions, was performed on the physical mixture. The temperature range used was 20–800 °C with a heating gradient of 10 °C/min. The analyses were performed in triplicate.

#### 2.2.5. X-Ray Powder Diffraction

X-Ray Powder Diffraction (XRPD) data were collected in the 2θ range 3–50° using a θ/θ diffractometer (X’Pert PRO PANalytical, Monza, MB, Italy, Cu Kα radiation) equipped with a real-time multiple strip detector. The incident beam optics included divergence and anti-scattering slits (0.5°) as well as a soller slit (0.02 rad). The diffracted beam passed through a soller slit (0.02 rad), a Ni filter and an antiscatter blade before arriving to the detector. Samples were gently ground and sprinkled on a zero-background sample holder (single crystal silicon). 

#### 2.2.6. X-ray Photoelectron Spectroscopy

The experiments by means of X-ray Photoelectron Spectroscopy (XPS) were carried out in ultra-high vacuum at a base pressure of 10^−9^ mbar. X-ray photoemission data were recorded with a VG Microtech Clam2 hemispherical electron analyzer operated at constant pass energy. Overview scans were acquired with a pass energy of 100 eV (2 eV resolution), while higher resolution spectra of single elements were acquired with a pass energy of 35 eV (0.7 eV resolution). X-ray photoemission was carried out with non-monochromatic Mg Kα photons (hν = 1253.6 eV) from a Vacuum Generators XR3 dual anode source operated at 15 kV, 18 mA. The Mg Kα satellite lines and the background (Shirley or linear type) were subtracted from the acquired spectra. The spectra were fitted with Voigt components for quantitative analysis. Following a standard procedure [37], from the peak areas and considering appropriate tabulated atomic sensitivity factors for the different peaks, a quantitative evaluation of the elements present at SLNas surface was then performed.

#### 2.2.7. Wettability

The wettability was determined on SLNas tablets obtained by direct compression of 100 mg sample using a hydraulic press (100 kg/cm^2^, 30 s, 40 mm diameter punches). The direct measurement of the contact angle, the tangent angle at the contact point between the water drop and the tablets, was carried out by the sessile drop method. In practice, drop profiles enlarged by a telescope were captured and the angles on both sides of the liquid drop were measured at time zero. The determinations were carried out in triplicate from three different batches.

#### 2.2.8. Drug Loading and Encapsulation Efficiency

SLNas drug loading, expressed as % *w*/*w*, was determined by HPLC analysis as previously described [34]. Briefly, analyses were carried out on a reversed-phase column (RP-18e, 125 × 4 mm, 5.0 μm) thermostatted at 30 °C and protected by a guard column (4.0 × 4.0, 5.0 μm) (Purospher, Merck, Darmstadt, Germany). The mobile phase consisted of methanol/0.02 M sodium acetate buffer (pH 5.0) using a gradient elution with a flow rate of 0.9 mL/min; the column eluates were monitored at 254 nm. Calibrators containing an internal standard (IS) and increasing RIF concentration were analyzed in triplicate; the peak-area ratio of RIF to IS was plotted against the RIF concentration to calculate the calibration curve by the method of least squares. For SLNas samples, exactly weighed aliquots, added with the IS solution and methanol, were heated at 50 °C for 30 min, filtered and analyzed by RP-HPLC (*n* = 3). RIF content was calculated by interpolation of the calibration curve and the encapsulation efficiency (EE) was estimated by comparing the percentage of RIF entrapped in SLNas (actual drug loading) compared to the initial RIF amount used in SLNas preparation (theoretical drug loading). Data were averaged on three determinations.

#### 2.2.9. Aerodynamic Diameter

SLNas aerodynamic diameter (d_a_), defined as the diameter of a sphere of unit density, was estimated by the following relationship: $$d_{a}=d_{v}\;\sqrt{\frac{\rho^{\mathrm{tapped}}}{{\chi \rho }^{0}}}$$ where d_v_ is d(BET), ρ^0^ is the unit density (of spherical calibration particles), ρ^tapped^ is the particle tapped density and χ is the dynamic shape factor, defined as circularity. 

#### 2.2.10. Respirability

In vitro powder respirability was assessed using Glass Twin Impinger (GTI) (Disa, Milan, Italy). SLNas samples were loaded (about 30 mg), equally distributed, into 3 capsules (V-Caps Capsugel, Morristown, NJ, USA) and aerosolized using a RS01 device (Plastiape, Lecco, Italy) with a flow rate of 55 L/min, capable of producing a pressure drop of 4 kPa over the inhaler. Considering RIF solubility, methanol was used as the collection liquid, of which 7 and 30 mL were inserted into the two GTI stages (stages A and B). The vacuum was applied for 4 s to obtain 4 L of air through the instrument during the experiment. After simulation, drug traces within the capsule/device (stage D) and all the other stages were washed with methanol and RIF was quantified spectrophotometrically (Lambda 3B Perkin-Elmer, Waltham, MA, USA) at the wavelength of 475 nm. Unlike SLNas size in stages D and A estimated >6 μm, the particles captured in stage B were considered as the fine particle fraction <6 μm. All the experiments were performed in triplicate.

#### 2.2.11. Lipid Corona Formation

In vitro lipid corona formation around SLNas was evaluated by size and surface charge analysis (1) and also by Energy Dispersive X-ray (EDX) analysis (2). 1) Size, PDI, and Z-potential of pure Curosurf and SLNas, before and after treatment with Curosurf, were determined using PCS. Curosurf was diluted at 1.5 mg/mL using a saline solution and added to 1 mg/mL of SLNas samples under magnetic stirring (200 rpm) at 37 °C for 1 h. The size was measured and compared with that measured on the samples suspended in saline solution. All the determinations were carried out in triplicate on properly MilliQ water diluted samples. 2) The elemental composition was determined by EDX analysis (INCA 350, Oxford Instruments, Abingdon, UK) coupled with ESEM (Quanta 200, FEI Company, Oxford Instruments, Abingdon, UK) for pure Curosurf and SLNas samples before and after treatment with Curosurf. Sample suspensions were dried on ESEM stubs before analyses. X-ray emission from Ka or Kb levels of the atoms C, O, Na, P, S, and Cl were recorded. The EDX spectra expressed as the plots of X-ray counts vs. element energy peak and semi-quantitative results, expressed as relative weight percentage of the elements, were recorded under the following conditions: low vacuum (0.70 Torr), accelerating voltage 20 kV, spot size 5, element detection limit ~0.05 wt%, spatial resolution 0.1 μm, total spectrum counts >250,000, accuracy within ±5% relative errors by reference to standards. The semi-quantitative analysis was determined by using the Φ(ρz) method [38]. The reported data were averaged over three determinations for each sample. 

#### 2.2.12. In Vitro Drug Release

In vitro RIF release from SLNas was performed in sink conditions by placing approximately 30 mg of SLNas powder suspended in 2 mL SLF alone or with Curosurf (1:1.5 SLNas:Curosurf ratio) within a dialysis membrane (cut off 12,000/14,000 D) and immediately soaked in 30 mL of SLF at 37 °C. The equipment was maintained in darkness under gently magnetic stirring to mimic the lung environment. Sample solutions were withdrawn at fixed time intervals and the initial volume restored. The solutions were analyzed spectrophotometrically (Lambda 3B Perkin-Elmer, Waltham, MA, USA) in triplicate at the wavelength of 475 nm.

#### 2.2.13. Cell Culture Assay

Murine alveolar macrophages MH-S cell line were cultured under mixed condition in RPMI 1640 medium supplemented with 2 mM l-glutamine, penicillin 100 UI/mL, 100 μg/mL streptomycin, and 10% FBS in T75 flasks at 37 °C and 5% CO_2_. For cytotoxicity and uptake experiments, cells were seeded in 24-well plates at a density of 120,000 cells/well or in 100 mm Petri dish at a density of 4,000,000 cells/Petri, respectively. 

#### 2.2.14. Cytotoxicity Test

MH-S cell line was seeded to evaluate the cytotoxicity of SLNas and pure RIF. Cells were incubated for 6 h with 1 mL of SLNas suspended in complete RPMI 1640 (0.125, 0.25, 0.5, and 1.0 mg/mL particle dose), vortexed for 1 min, and sonicated in sonic bath at 37 °C for 3 min. Regarding RIF, cells were incubated with 10 µL RIF solution in DMSO at a drug concentration range of 10–200 µg/mL for 24 h as recommended for antibiotic toxicity on macrophages [39] (10 µL DMSO as the control). 

Cell viability was estimated by MTT test using a multiplate reader (TecanGenios Pro with Magellan 6 software, MTX Lab Systems, Bradenton, FL, USA) at an optical density of 535 nm to monitor the formazan formation by metabolically viable cells. The test was performed in triplicate and the results were expressed as percentage of cell survival compared with the untreated cells.

#### 2.2.15. Intracellular RIF Determination

MH-S cells were plated in a 100 mm Petri dish and incubated with SLNas suspensions at 0.25 mg/mL for 0.5, 1, and 3 h. After incubation, adherent cells were detached from the plate by addition of trypsin-EDTA and joined to the suspended cell fraction. After washing with PBS, RIF embedded inside cells was extracted by treatment with methanol (1 mL). Drug concentration was determined spectrophotometrically at 475 nm and expressed as RIF percentage on the specific drug loading level of each sample batch. The reported data were averaged on three determinations of three different batches.

#### 2.2.16. MR Inhibition Study

MH-S cells were seeded in a 100 mm Petri dish and treated for 2 h with 50 mM d-mannose to saturate the mannose receptor on macrophage membrane. Subsequently, cells were incubated with SLNas (0.25 mg/mL) for 0.5, 1, and 3 h. Then, cells were handled as described in the previous section for the intracellular RIF determination. Reported data were averaged on three determinations of three different batches.

#### 2.2.17. Intracellular RIF Determination after Lipid Corona Formation

MH-S cells were seeded in a 100 mm Petri dish with cell medium without FBS and treated with SLNas previously suspended in cell medium (0.25 mg/mL) with Curosurf at 1:1.5 ratio (SLNas:Curosurf) and sonicated in the ultrasonic bath for 10 min at 37 °C. The cells were incubated with the mixture for 0.5, 1, and 3 h. Then, cells were handled as described in the previous section for the intracellular RIF determination. Reported data were averaged on three determinations of three different batches.

#### 2.2.18. Statistical Analysis

Data obtained were evaluated statistically using one-way analysis of variance (ANOVA). Significance was indicated by *p* < 0.05 (* *p* < 0.05; ** *p* < 0.01; *** *p* < 0.005).

## 3. Results and Discussion

Based on the satisfactory results already achieved by anchoring newly synthesized mannosylated surfactants to SLNas surface [5], the purpose of this study was to investigate if the pulmonary surfactant lining the alveolar surface could possibly form, due to its high content of phospholipids, a lipid corona around the nanoparticles that might nullify alveolar macrophage (AM) active targeting and make SLNas functionalization unnecessary. The investigation involved both mannosylated SLNas (SLNas/MS) and SLNas functionalized by MS in a blend with sodium taurocholate (SLNas/MS-ST) in order to modulate the mannose concentration on the nanoparticle surface. Indeed, ST itself can play a key role, since an activity as opener of tight junctions between alveolar epithelial cells has been described [40], although the definitive mechanism has not been fully elucidated yet. In addition, it is also noteworthy that mannose receptors (MR) bear a cystein-rich domain that recognizes sulfonic group. SLNas obtained by using ST (SLNas/ST) or F127 (SLNas/F127), a non-ionic surfactant devoid of groups able to bind MR, were evaluated as the controls.

### 3.1. SLNas Physico-Chemical Properties

The study of inhaled particulate drug carriers for AM active targeting, intended to increase particle specificity for macrophages and internalization potential over the passive targeting, must not disregard the investigation of the physico-chemical features required to both reach alveolar region and be phagocytized by AM. Therefore, the obtained SLNas samples were characterized for morphology, size, surface charge, shape factor, physical state, wettability, surface properties, density, flowability, drug loading, drug release, and respirability performance. 

It is well known that the efficiency of an inhaled powder formulation to settle into the deep lung depends upon particle geometrical parameters (shape, morphology, and size) that affect the powder respirability as well as macrophage uptake [6]. The most commonly used particles for DPI device exhibit irregular shapes. In vitro inhalation studies indicated that elongated [41,42], needle-like [43], porous and wrinkle [44], as well as thin flaky shapes [45] can improve lung deposition properties owing to reduced particle interactions [46]. Concerning macrophage phagocytosis, particle shape has shown to affect cell uptake mechanism or rate, with roundness being a preferred characteristic compared to elongated or filamentous shapes [6,47,48]. All the obtained SLNas samples exhibited rounded micro-aggregates of nanoparticles having an irregular shape with tendency to roundness, as indicated by circularity values from 0.6 to 0.83, with 1 indicating perfect roundness (Figure 1, Table 1).

Particle size of the samples, measured by PCS and expressed as the mean diameter of the main class (>85%), was within the range of about 300 to 750 nm, where the larger size was shown for SLNas/MS and SLNas/ST compared with that for SLNas/MS-ST and SLNas/F127 (*p* < 0.05). The size measured on the samples in their solid state by means of the BET technique, d(BET), aimed to simulate the therapeutic use of powdered drug by means of a DPI device. The d(BET) values were within the range from about 700 nm, for SLNas/MS, to 1100 nm for all the other samples (*p* < 0.05). This size is considered proper for the deposition onto alveolar region and AM endocytosis [6]. The related PDI values ranged from 0.3 to 0.6, owing to the presence of minor populations in the range of 2–5 μm (<15%), owing to a multimodal size distribution of the particle aggregates. The surface charge of all the samples was negative (from −15 to −55 mV). Considering that SLNas include the same lipid matrix, the largest negative values were reasonably imparted by the negatively charged MS and ST unlike what occurred with the non-ionic surfactant F127. Extent and rate of macrophage uptake are known to be directly related to particle net charge magnitude, with negatively charged surface being more physiologically compatible than the positive ones [6]. Moreover, unlike particles bearing positive charges, negative surface charges showed to favor particle localization within lysosomes, in which *Mycobacterium tuberculosis* survives [49,50]. As regards SLNas density features, true density values ranged from 1.147 to 1.247 g/cm^3^, bulk density from 0.031 to 0.147 g/cm^3^, and the respective tapped density values from 0.038 to 0.29 g/cm^3^; SLNas/F127 exhibited the highest values of both bulk and tapped density among the samples. The increased tapped density compared to the bulk density values indicates the presence of inter-particle void spaces. Tapped density represents a pivotal parameter for both flowability and aerodynamic diameter required for a respirable powder. Indeed, particles having low tapped density can efficiently aerosolize from a DPI, resulting in a high respirable fraction of inhaled therapeutics [51]. Flowability values, expressed as Carr’s Index, were affected by SLNas surface decoration; in particular, MS provided excellent flow properties also in the presence of ST (Carr’s Index <10) [52]. Conversely, SLNas/ST and SLNas/F127 flowability resulted fair and very poor, respectively. The flow properties of a powder as well as its de-aggregation and aerosol performance can also be influenced by inter-particulate interactions related to the specific surface area, i.e., the surface area per unit mass. Such a parameter depends on both particle size and surface roughness of the powder [41,53]. The average values for BET specific surface area of the SLNas samples were found to be 5–7 m^2^/g.

Aerodynamic diameters were calculated for all the SLNas samples on the basis of their shape, BET size, and density characteristics and found to be in the range of 210–676 nm. Despite the values should be higher than 0.5 μm to promote particle deposition onto the alveolar epithelium [46,54], the analysis of the respirability performance, which reflects also the degree of particle cohesiveness and de-aggregation, provided more reliable results. The percentage values of Emitted Dose (ED) and Fine Particle Fraction (FPF) were measured by GTI (Table 2) to define the suitability for the powder to be discharged during the air actuation by DPI and the fraction deposited into alveolar region. Mannosylated samples (SLNas/MS and SLNas/MS-ST) as well as SLNas/ST showed ED values >80%, that is the threshold required by the European Pharmacopoeia [55] and FPF data around or >40%, indicating adequate respirability performance for antibiotics administered by DPI [56]. Only SLNas/F127 did not comply with powder respirability requirements (about 70% ED and 12% FPF); this was probably due to its very poor flowability and/or lower surface charge, which led to greater cohesiveness and lower de-aggregation ability compared with the other samples.

Particulate carriers must exhibit adequate respirability characteristics, but they also must ensure an adequate drug payload within them to achieve a therapeutic effect with a feasible dose inside the DPI. In addition, the carriers have to possess the ability to retain the drug before their phagocytosis by AM [57]. The obtained RIF loading levels and EE values were about 9% and 35%, respectively, without significant differences among the samples (*p* > 0.05) (Table 3). 

By considering that RIF was stable under the adopted preparation conditions, as previously demonstrated [34], EE values indicate that RIF leached partly from the lipid matrix into the continuous phase of the emulsion during the preparation process or dialysis phase, as monitored in preliminary studies. The loaded RIF was found in the amorphous state within all SLNas matrices, as demonstrated by DSC analyses (Figure 2). 

Indeed, RIF characteristic endotherm at about 260 °C (Figure 2f) was present only in the physical mixture (Figure 2e). The thermal events of PA and CM, ranging from 50 to 70 °C, were observed in both SLNas and physical mixture thermograms and no modification occurred during the second heating run (data not shown). Thermogravimetric profiles revealed mass losses related to the degradation of SLNas components in the range of 200–600 °C. No mass losses around 100 °C, attributable to dehydration processes, were observed in the samples except for SLNas/ST, which showed a mass loss of about 4% (Figure 2c). RIF within SLNas lipid matrices was confirmed to be in an amorphous state by XRPD patterns. Unlike the physical mixture showing RIF characteristic peaks at 9.9°, 11.1°, and 19.9° 2θ, even if with decreased peak intensity, all diffractograms exhibited the typical peaks of the matrix components except for RIF peaks, regardless of the sample (Figure 3).

XPS analysis and wettability measurements were carried out to determine the chemical nature provided by the functionalizing moieties on the SLNas surface. Concerning XPS analysis, a technique for analyzing 5 nm-thick surface layer of the sample, the obtained data indicate the quantitative levels of the elements present on the sample surface (Table 4).

Carbon (C_1s_), oxygen (O_1s_), and nitrogen (N_1s_), this latter belonging to MS, ST, and RIF, were found on the surface of all the samples, whereas sulphur (S_2p_) and sodium (Na_1s_) were detected only in SLNas samples prepared using ST as surfactant or co-surfactant agent. The lowest levels of nitrogen atoms were detected in SLNas/F127 owing to the only contribution of RIF close to the sample surface. Consequently, the highest nitrogen levels exhibited by the other samples are reasonably attributable to the functionalizing/surfactant agents MS and ST located on the particle surface. 

As regards wettability, contact angles less than 90° were measured for all SLNas samples revealing that wetting of the surface was favorable (Table 4). The surface wettability is conceivably attributable to the hydrophilicity conferred by the surfactants to the hydrophobic nanoparticle lipid matrix. The weakest water-wet properties were given by SLNas/F127 in accordance with the values assigned to the brush-like conformation adopted on hydrophobic surfaces by the PEO chains of the surfactant [58,59].

### 3.2. Lipid Corona Formation upon SLNas Contact with Pulmonary Surfactant

Once nanoparticles administered in a dry powder form reach pulmonary alveoli, pulmonary surfactant is the first lung component they encounter as a monolayer or oligolamellar layer at the air-liquid interface of the fluid lining alveoli [60]. Pulmonary surfactant is also present in the case of human respiratory infections even if with reduced availability of lipids and proteins [12]. In order to evaluate the possible occurrence of interactions between SLNas samples and pulmonary surfactant, SLNas were treated with Curosurf, a commercial substitute of the natural pulmonary surfactant, and analyzed for size, PDI, zeta potential, and elemental composition. Moreover, the effect of the lipid corona around the nanoparticles on in vitro drug release was assessed.

The interactions between pulmonary surfactant and nanoparticles can be assessed by means of changes in size and zeta potential of the nanoparticles. After the treatment with Curosurf, all the samples increased significantly (*p* ≤ 0.05) in size by 1.5 to 5-fold (Table 5), except SLNas/F127 (*p* > 0.05). This finding could be presumably related to the insufficient affinity of Curosurf for F127 functional groups as reported by other authors [23]. 

As regards zeta potential, the negative value exhibited by Curosurf (−32 ± 3 mV) was not expected to modify significantly the negative surface charge of SLNas samples and this impaired the contribution of such a measurement to corona formation assessment. 

The key role played by the phospholipid fraction in the pulmonary surfactant may justify the use of the EDX analysis focused on phosphorous element detection. The elemental analysis performed by EDX by means of the single point method provided the semi-quantitative composition of each SLNas sample before and after treatment with Curosurf. Moreover, each corresponding ESEM image showed, for all the samples treated with Curosurf, the presence of distinct particles surrounded by a poorly dense layer ascribable to the pulmonary surfactant. Only few aggregates were observed (Figure 4). The presence, in the samples treated with Curosurf, of phosphorous belonging exclusively to phospholipids as well as sodium chloride contained in Curosurf saline solution was reasonably due to the formation of a phospholipid corona around the particles. Phosphorous was detectable in all the SLNas samples, without significant differences among them (*p* < 0.05), with the exception of SLNas/F127 (level below the detection limit of <0.05%) (Table 6). Unlike sodium and chlorine X-ray emissions due to both material impurities and ST, phosphorous levels were below the detection limit in all the untreated samples (data not shown). Hence, EDX analysis further confirms the lipid corona formation due to interactions between the pulmonary surfactant substitute and the SLNas surface decorated with both MS and ST.

### 3.3. Effect of Pulmonary Surfactant on Drug Release and Macrophage Activity

Lung surfactant, especially its phospholipid fraction, may impact drug release before AM internalization, as well as particle agglomeration, and cellular uptake due to lipid corona formation on the SLNas surface [61]. SLNas designed to be taken up by AM should be able to retain most of the active principle upon contact with lung fluids until its entry into AM. Based on that, in vitro drug release from SLNas samples was evaluated in SLF containing or not Curosurf (Figure 5). 

Drug release in pure SLF was gradual from the lipid matrix of mannosylated SLNas (SLNas/MS and SLNas/MS-ST; Figure 5a,b) and reached RIF percentages of about 40% in 3 h. In contrast, about 60% RIF was released from the control samples SLNas/ST and SLNas/F127 (Figure 5c,d). Drug retaining capability exhibited by mannosylated samples could be attributed to a closer settlement of the functionalizing agent lipophilic tale within the particle lipid phase which may generate a firmer SLNas matrix as observed in a previously published research [5]. Therefore drug spreading over the lung fluid before AM uptake should not occur for mannosylated SLNas, since 50–75% of inhaled particles should be taken up by AM in 2–3 h [47]. RIF release in SLF containing Curosurf decreased by 20–25% (*p* < 0.05) for all the samples, except for SLNas/F127, that is the only sample that proved to fail in interacting with the pulmonary surfactant substitute. The observed improvement in drug retention, attributable to an additional resistance to drug diffusion provided by lipid corona layer, could be considered as an advantage as it leads to a greater amount of drug available within the lipid matrix to be transported into AM. After AM phagocytosis, lipid materials are known to be degraded inside macrophages by lipolytic enzymes, allowing drug intracellular leakage and antibacterial activity [62].

Besides the conflicting results concerning the effects of protein/lipid corona on biological interactions between nanoparticles and cells, it is reasonable to expect that the activity generated by the surface chemistry of particles coated by the pulmonary surfactant layer might undergo some changes upon contact with AM membrane and its MR. Consequently, the effect of the pulmonary surfactant layer on SLNas ability to be phagocytosed with or without impact on MR interactions was investigated on MH-S cell line, referred as an in vitro model similar to human primary cells [63]. All the samples and pure RIF were preliminarily subjected to cytotoxicity test according to the incubation time.

SLNas cytotoxicity was investigated at increasingly sample amounts (from 0.125 to 1 mg/mL) and a significant dose-dependent cell cytotoxicity was found for the mannosylated SLNas samples starting from the dose of 0.5 mg/mL (*p* < 0.05) (Figure 6a). 

Since excipient concentrations and RIF loading levels are recognized as safe (non-cytotoxic) in the obtained SLNas (Figure 6b), the dose-dependent cell apoptosis could be reasonably attributed to the increase of particle amounts [64] or particle clustering phenomena [65,66]. With respect to SLNas/F127, such an effect may have been precluded by F127 surfactant recognized to be capable of inhibiting nanoparticle cell adhesion [67].

Based upon the cytotoxicity results, the study of SLNas macrophage internalization was performed on 0.25 mg/mL SLNas dose, corresponding to about 22.5 μg/mL RIF for all the samples, that is higher than the minimum inhibitory concentration against *Mycobacterium tuberculosis* strain (1 μg/mL) [68]. Intracellular RIF concentrations were determined on SLNas samples up to an incubation time of 3 h that is considered consistent with both AM phagocytosis rate [47,69] and the occurrence of intracellular RIF [5]. The internalization data were compared with those obtained following both MR inhibition and pulmonary surfactant co-treatment (Figure 7). 

Fast and complete drug translocation within AM was achieved by cell exposure to SLNas/MS and reached 100% intramacrophagic RIF at 0.5 h incubation (Figure 7a). This maximum drug concentration was maintained intracellularly until drug degradation occurs, leading to a concentration decrease by one-half at 3 h incubation (*p* < 0.05). Conversely, SLNas/MS-ST sample behaved as the controls SLNas/ST and SLNas/F127 and showed RIF translocation values less than 50%, regardless of the incubation time (Figure 7b–d). The greatest internalization ability of SLNas/MS is therefore the consequence of SLNas mannose recognition by MR, which are highly expressed on MH-S cell line [70]. By contrast, ST seemed to not contribute to AM internalization through mechanisms of active targeting. Comparison between drug translocation efficiency provided by SLNas/MS and that by SLNas/MS-ST therefore gives an indication of the key role played by mannose density on nanoparticle surface [71]. MR-mediated uptake by macrophages was investigated by cell pre-treatment with d-mannose to saturate the corresponding receptors [5,72,73]. MR inhibition resulted in a relevant decrease (about 50%) of intracellular RIF only by cell exposure to SLNas/MS at 0.5 h (*p* < 0.005) and at 1 h (*p* < 0.01), before drug degradation occurrence at 3 h incubation (Figure 7a), confirming the considerable contribution from the active targeting process. Otherwise, RIF transport across macrophage membrane was only slightly affected (10% intracellular RIF decrease) by MR inhibition upon cell contact with SLNas/MS-ST (*p* < 0.05 at 1 and 3 h; Figure 7b) and SLNas/ST (*p* < 0.05 at 3 h; Figure 7c), suggesting a marginal contribution from MR-mediated cell uptake. Insignificant changes in RIF translocation were detected for SLNas/F127 (*p* > 0.05; Figure 7d), regardless of the incubation time. It follows that a passive targeting could be assumed as the main mechanism for the macrophage uptake of these latest SLNas samples. 

Concerning the effect of pulmonary surfactant on SLNas cell uptake, the interactions between mannose groups on SLNas/MS surface and macrophage MR were negligibly impaired by the presence of the pulmonary surfactant substitute, regardless of the incubation time (*p* > 0.05) (Figure 7a). This finding suggests that the adsorbed lipid corona layer does not completely mask the functional groups on nanoparticle surface as observed also by other authors [74]. Corona formation appeared instead to promote macrophage internalization when passive targeting regulates the process, i.e., for SLNas/MS-ST and SLNas/ST samples, leading in general to increased RIF intracellular concentrations (from about 10 to 20%) (*p* < 0.05; Figure 7b,c). Different mechanisms could be hypothesized for an increased intracellular uptake generated by pulmonary surfactant coating on nanoparticles, including recognition promotion by macrophages and hydrophobic interactions or fusion between nanoparticle surface and macrophage membrane similarly to the process involving liposomes [31]. The lack of a significant uptake modification provided by pulmonary surfactant on SLNas/F127 (*p* > 0.05) proved further the absence of lipid corona. 

## 4. Conclusions

In previous researches, SLNas functionalized by a mannosylated surfactant for tuberculosis inhalation therapy were implemented in terms of production technique and active targeting to alveolar macrophages. The successful results thus achieved encouraged the present investigation that deals with the effect of pulmonary surfactant on drug translocation within alveolar macrophages. Even though our findings demonstrated that mannose functional groups interact with pulmonary surfactant, their density on SLNas surface influences both drug receptor-mediated transport into macrophages and the intramacrophagic transport promoted by the lipid corona. These results highlight the need for specific studies according to each modification introduced on the surface of inhaled carriers designed to be taken up by alveolar macrophages. Further studies to address in vivo drug targeting potentiality and bio-distribution on animal models are ongoing.

## Figures and Tables

**Figure 1 pharmaceutics-11-00508-f001:**
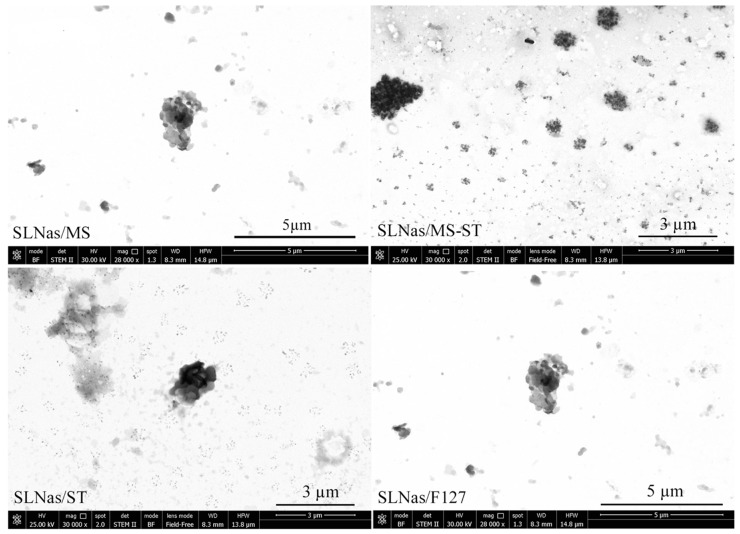
Transmission electron microscopy images of the SLNas samples.

**Figure 2 pharmaceutics-11-00508-f002:**
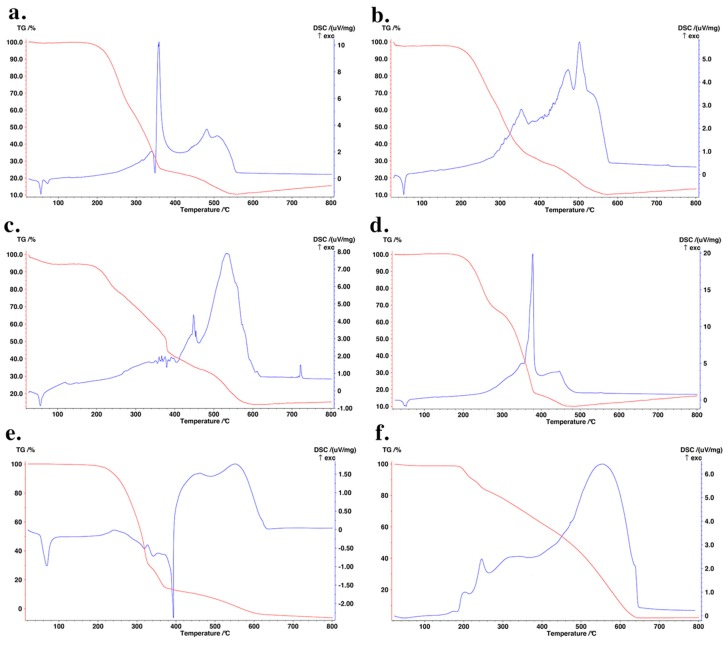
Simultaneous thermogravimetry (red line) and differential scanning calorimetry (blue line) of SLNas/MS (**a**), SLNas/MS-ST (**b**), SLNas/ST (**c**), SLNas/F127 (**d**), physical mixture (**e**), and rifampicin (**f**).

**Figure 3 pharmaceutics-11-00508-f003:**
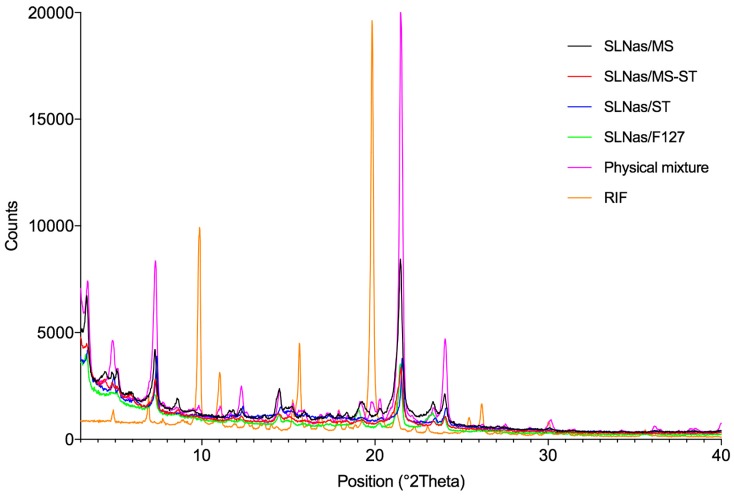
XRPD profiles of the SLNas samples in comparison with the physical mixture of SLNas components as well as pure rifampicin.

**Figure 4 pharmaceutics-11-00508-f004:**
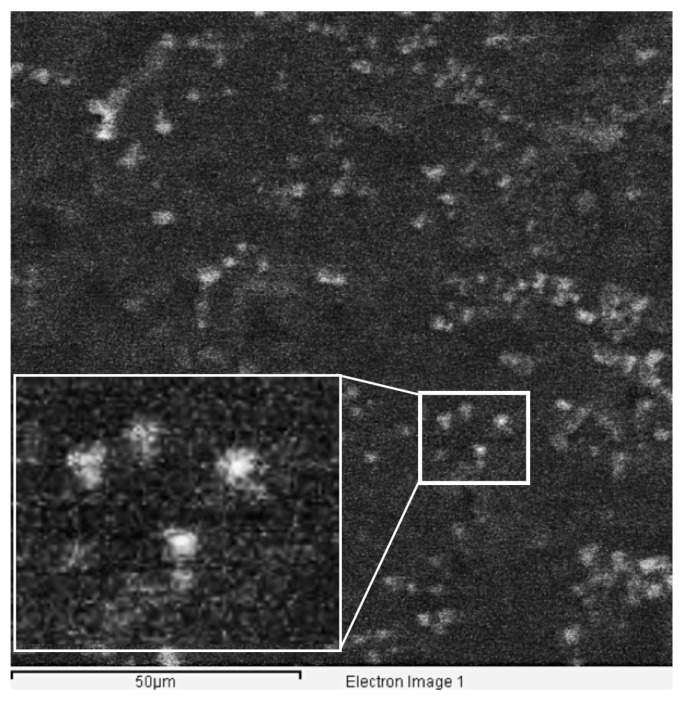
Representative ESEM image with an enlarged frame of a SLNas/MS sample treated with Curosurf.

**Figure 5 pharmaceutics-11-00508-f005:**
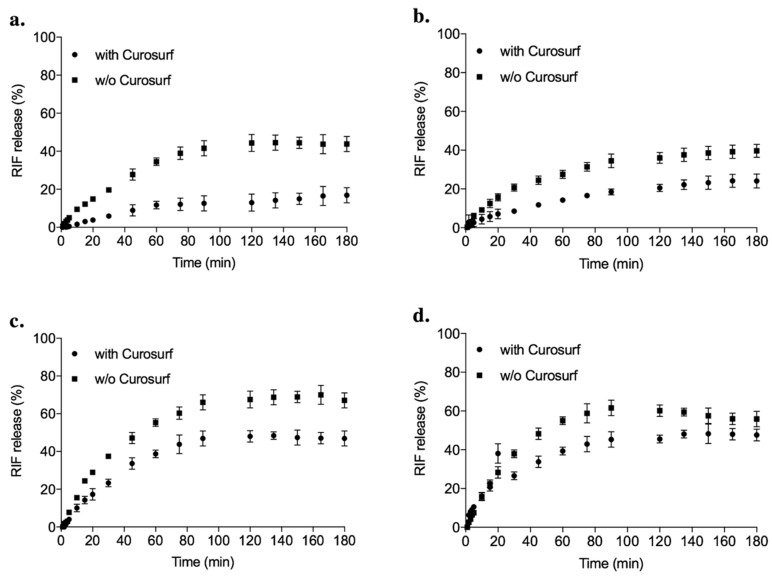
Rifampicin release from SLNas/MS (**a**), SLNas/MS-ST (**b**), SLNas/ST (**c**), and SLNas/F127 (**d**) in simulated lung fluid with or without (w/o) the pulmonary surfactant substitute (Curosurf).

**Figure 6 pharmaceutics-11-00508-f006:**
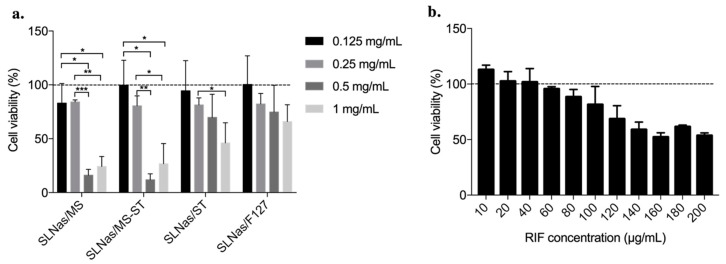
Cytotoxicity of SLNas samples at 6 h incubation time (**a**) and pure rifampicin at 24 h incubation time (**b**) determined on MH-S cell line. Dotted line as the control. Significance was indicated by * *p* <0.05, ** *p* < 0.01, *** *p* < 0.005.

**Figure 7 pharmaceutics-11-00508-f007:**
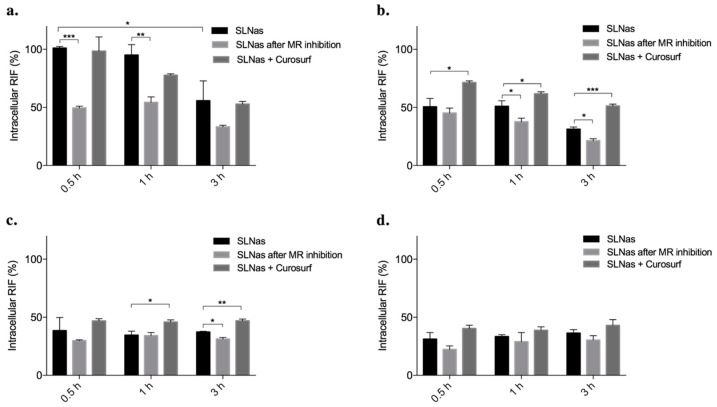
Intramacrophagic rifampicin percentages upon cell exposure to SLNas samples in comparison with those determined following both cell mannose receptor inhibition and co-treatment with pulmonary surfactant substitute (Curosurf). SLNas/MS (**a**), SLNas/MS-ST (**b**), SLNas/ST (**c**), and SLNas/F127 (**d**). Significance was indicated by * *p* < 0.05, ** *p* < 0.01, *** *p* < 0.005.

**Table 1 pharmaceutics-11-00508-t001:** Physical characteristics of the SLNas samples (mean values ± SD).

Physical Characteristics	SLNas/MS	SLNas/MS-ST	SLNas/ST	SLNas/F127
Circularity	0.6 ± 0.1 ^a^	0.83 ± 0.03 ^b^	0.74 ± 0.07 ^ab^	0.75 ± 0.06 ^ab^
Size (nm)	740 ± 85 ^a^	309 ± 30 ^b^	668 ± 25 ^a^	408 ± 57 ^b^
PDI	0.60 ± 0.05	0.30 ± 0.02	0.46 ± 0.05 ^a^	0.46 ± 0.05 ^a^
Z potential (mV)	−35.2 ± 0.1	−40.5 ± 0.9	−55 ± 2	−15 ± 0.1
d(BET) (nm)	730 ± 10	900 ± 20	1020 ± 50 ^a^	1090 ± 20 ^a^
ρ true (g/cm^3^)	1.147 ± 0.001	1.1939 ± 0.0008	1.247 ± 0.002	1.1791 ± 0.0008
ρ bulk (g/cm^3^)	0.048 ± 0.001	0.079 ± 0.001	0.031 ± 0.000	0.147 ± 0.005
ρ tapped (g/cm^3^)	0.052 ± 0.001 ^a^	0.086 ± 0.001	0.038 ± 0.000 ^a^	0.29 ± 0.02
Carr’s Index	9 ± 2 ^a^	8 ± 2 ^a^	17 ± 1 ^a^	50 ± 8
BET area (m^2^/g)	7.2 ± 0.10	5.6 ± 0.10	4.7 ± 0.20 ^a^	4.67 ± 0.09 ^a^
d_a_ (nm)	210	288	231	676

^ab^ Among columns, means that have no superscript in common are significantly different from each other (*p* < 0.05).

**Table 2 pharmaceutics-11-00508-t002:** Aerodynamic parameters of the SLNas samples provided by Glass Twin Impinger: Emitted Dose (ED) and Fine Particle Fraction (FPF). Mean values ± SD.

SLNas Samples	ED (%)	FPF (%)
SLNas/MS	87 ± 4 ^a^	38 ± 5 ^a^
SLNas/MS-ST	84 ± 2 ^ab^	41 ± 5 ^a^
SLNas/ST	83 ± 1 ^ab^	53 ± 4
SLNas/F127	72 ± 8 ^b^	11.77 ± 0.01

^ab^ Among lines, means that have no superscript in common are significantly different from each other (*p* < 0.05).

**Table 3 pharmaceutics-11-00508-t003:** Drug loading (DL) and encapsulation efficiency (EE) values of the SLNas samples. Mean values ± SD.

SLNas Samples	DL (%)	EE (%)
SLNas/MS	9.2 ± 0.2	36.8 ± 0.9
SLNas/MS-ST	8.7 ± 0.2	34.9 ± 0.9
SLNas/ST	8.9 ± 0.3	36 ± 1
SLNas/F127	8.4 ± 0.8	34 ± 3

**Table 4 pharmaceutics-11-00508-t004:** SLNas surface elemental composition (relative percentage of C, N, O, S, and Na) evaluated by XPS analysis and wettability expressed in terms of contact angle (θ).

SLNas Samples	C	O	N	S	Na	θ (deg)
SLNas/MS	71	25.5	3.3	0.0	0.0	51 ± 5
SLNas/MS-ST	82	15.0	2.3	0.3	0.2	32 ± 4
SLNas/ST	89	8.4	1.1	0.7	0.6	37 ± 8
SLNas/F127	86	13.3	0.6	0.0	0.0	79 ± 2

**Table 5 pharmaceutics-11-00508-t005:** Dimensional values (size and PDI) of the SLNas samples before and after treatment with Curosurf. Mean values ± SD.

SLNas Samples	In Saline Solution		In Saline Solution with Curosurf		Fold Increase in Size
	Size (nm)	PDI	Size (nm)	PDI	
SLNas/MS	962 ± 286	0.45 ± 0.20	1611 ± 272	0.4 ± 0.2	1.67
SLNas/MS-ST	356 ± 64	0.29 ± 0.13	1839 ± 90	0.22 ± 0.01	5.17
SLNas/ST	284 ± 36	0.27 ± 0.04	884 ± 158	0.31 ± 0.02	3.11
SLNas/F127	551 ± 6 ^a^	0.68 ± 0.05	837 ± 272 ^a^	0.5 ± 0.3	1.52

^a^ Among lines, superscript indicates no significant difference among the size without Curosurf and that with Curosurf from each sample (*p* > 0.05).

**Table 6 pharmaceutics-11-00508-t006:** Elemental composition by EDX analysis of pure Curosurf and of SLNas samples treated with Curosurf (relative weight %, mean values ± SD).

Samples	C	O	Na	P	S	Cl
Curosurf	71 ± 2	22 ± 2	1.1 ± 0.1	4.6 ± 0.6	0.04 ± 0.04	1.6 ± 0.2
SLNas/MS	85 ± 3	8 ± 1	3 ± 2	0.3 ± 0.1	0.03 ± 0.04	4 ± 2
SLNas/MS-ST	83 ± 3	8 ± 3	4 ± 3	0.3 ± 0.2	0.01 ± 0.01	5 ± 3
SLNas/ST	91 ± 1	8 ± 1	0.6 ± 0.2	0.26 ± 0.06	0.02 ± 0.02	0.7 ± 0.2
SLNas/F127	94.39 ± 0.05	3.94 ± 0.3	0.7 ± 0.1	0.02 ± 0.01	0.01 ± 0.00	0.9 ± 0.2

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
