# Peer review of "The Impact of Lipid Corona on Rifampicin Intramacrophagic Transport Using Inhaled Solid Lipid Nanoparticles Surface-Decorated with a Mannosylated Surfactant"

_pharmaceutics, 2019, doi:10.3390/pharmaceutics11100508_

Round 1

Reviewer 1 Report

This is an interesting manuscript where the authors determined the effect of the formulation of NP on their characteristics and drug delivery potential. I find it surprising that the authors have not added any in vitro efficacy data against TB to show that changes that cause better delivery to lung macrophages results in higher efficacy against TB bacteria.  A drug delivery system has to target a drug to the site of infection and also has to reach the site of the organism within a cell. In addition a drug’s efficacy within a cell may be adversely effected by its deactivation and excretion processes.  I think the authors should approach a collaborator with a TB model to determine the efficacy of the formulations that have been developed.

The authors may also like to take the following comments into consideration:

The English needs addressing in this manuscript – I have given some examples below:

Line 108 ‘mix by ultrasounds’ – should be mixed by ultrasonification

Line 109 ‘obtained’ should be ‘resulting’

Line 328 ‘Concerning SLNas density features’ should be rewritten as this is not a  good way  to start a sentence.  

Line 455 ‘release extents’ should be rewritten

Line 479 ‘attributed to a certain particle clustering’ should be rewritten

Tables 1, 2 and 5 Are there any significant differences in the characteristics of the NPs? Is the lack of statistical differences mean that the NP are similar

Are there any significant different in Figs 4 data?

Reviewer 2 Report

The objective of this manuscript to study on the effect of the corona on mannosylated targetting is an interesting subject and which has to be appreciated. But, I believe some more intensive work is needed to confirm these finding presented in the current manuscript.

The major concerns are:

In figure 3. Is the physical mixture containing RIF? what is the concentration of RIF in the physical mixture?. I am not able to see prominent RIF peak in the physical mixture. Is the concentration of RIF is low are the physical mixture itself forming any SLN? Table 5. The curosurf coated particles size has been increased. Is this happened because of the aggregation of just coating? Please do some more characterization like TEM where you can compare only SLNs and coated SLNs.  Figure 4. Is the initial load of the RIF equal in curosurf treated and untreated sample. Does some of the RIF were leaked during coating itself. Please comment on it. Line 470-472: The SLN did not show dose-dependent cell viability. It is showing dose-dependent cytotoxicity. The sentence is misleading the audience thus I request you to reframe the sentence. Line 477-480 and Figure 5. Please give some details about MIC of RIF on Mycobacterium tuberculosis and MIC of SLN then compare with the cytotoxicity results. Figure 6 and Uptake studies: Please check the uptake of these SLNas-MS and coated particles with the cells which have no MR (any normal epithelial cells). I believe if there is no significant uptake of both of these particles on normal cells then the hypothesis by the authors will be validated.

Reviewer 3 Report

The present manuscript intended to investigate the interactions between pulmonary surfactant and SLNas functionalized with a mannosylated surfactant and the effect of such interactions on internalization ability by murine alveolar macrophages. The topic is interesting and of scientific significance. However, although the data concerning the characterization of physiochemical properties were very clearly and coherently, the key data focusing on the interactions between mannosylated SLNas and pulmonary surfactant and the effect of such interactions on internalization of SLNas were not sufficient to draw a solid conclusion.

The major concerns.

(1) Concerning the study of lipid corona formation: Two methods (XPS and EDX) were used to evaluate the SLNas surface elemental composition, please state the reason. The chemical composition of lipid corona was only studied using EDX analysis. However, the sample treatment protocol for the EDX analysis was not specified and it was not clear if the EDX result was arisen from the lipid corona formation or due to hydration effect. Compared to the XPS data of SLNas/MS, the O content in the SLNas/MS treated with Curosurf from EDX analysis decreased from 25.5% to 8% whereas the Na content increased from 0 to 3%, does this mean the main chemical composition of lipid corona was attributable to NaCl, please discuss in more detail.  

(2) Concerning macrophage uptake: Macrophage uptake is particle size dependent and the particles with size of 1 - 6 mm are more susceptible to the uptake by alveolar macrophages than nanoparticles. So it is not clear why the particle size of SLNas was not designed at size range between 1 mm and 6 m In Fig 6, both SLNas/MS-ST and SLNas/ST with lipid corona formation resulted in significantly higher uptake by macrophage than those under MR inhibition, even SLNas/MS-ST, suggesting the lipid corona formation significantly improved macrophage uptake. Considering that there existed significant differences between particle size of SLNas/MS and SLNas/MS-ST or SLNas/ST, it is not clear whether the improved internalization ability of SLNas/MS with lipid corona formation was resulted from the effect of lipid corona and/or particle size dependent passive targeting rather than active targeting of mannosylation.

(3) As for the optimal aerodynamic diameters for deposition onto the alveolar epithelium: In this manuscript, the aerodynamic diameters of DPI were set in the range of 210-676 nm, obviously, such aerodynamic diameters are far from ideal since particles with aerodynamic diameters less than about 0.5 μm have decreased amount of impaction in the lung and are frequently exhaled. The statement in Line 344-345 “Aerodynamic diameters were in the range of 210-676 nm values that are considered proper to promote particle deposition onto the alveolar epithelium” was not true.

Minor concerns:

(4) SLNas preparation was based on freeze-drying, which is not a      suitable process for the production of inhalable particles. Although freeze-dried particles might be used for the aim of the present study, the evaluation of aerodynamic diameter and respirability of SLNas is not necessary since DPI is not the choice of dosage form for the inhalation delivery of the present SLNas.

(5) Glass Twin Impinger was used for the evaluation of respirability and the cut-off aerodynamic diameter for the FPF was 6.4 μm, which is poorly correlated with alveolar deposition.

(6) The DSC and TGA data in Fig 2 were not informative and it was not clear why the heating temperature was set as high as 800 degrees.

(7) XRPD assay appeared not be sensitive to determine whether the samples were in crystalline or amorphous state since the peak in the physical mixture was fairly low.

Round 2

Reviewer 1 Report

This is an improved manuscript but I have the following suggestions the authors may like to consider.

Abstract

Line 28: ‘Our studies demonstrated the lipid corona formation around SLNas in the presence of Curosurf, a commercial substitute of the natural pulmonary surfactant substitute’.  I do not understand this sentence.  Do the authors mean that studies showed a lipid corona formed around the NPs?

Line 29 ‘Curosurf interactions with SLNas surface decorated solely with the mannose surfactant alone led, on one side, to improved drug retention within SLNas before being taken up by AM phagocytosis takes place’. I do not understand this sentence either. Do the authors mean that using curosuf in the formulation prevented drug leakage from the NP so that when AM phagocytosed the NP they took up more drug than compared to NP without curosurf?  And that higher drug loaded was not associated with active targeting via the mannose receptor?

Line 147 ‘On the other hand’ I would delete this text from the sentence so that it reads “The tapped …”

I appreciate that significant differences are shown in the text but I think that significant differences should also be shown on the data in Tables and Figures as this help researchers interpret the data being shown.  I would strongly recommend that the author show the significant differences in Tables 1, 2 and 5 using traditional methods e.g. *p < 0.05 compared to control NP, ap < 0,05 treatment a vs treatment b

Line 402 ‘Besides the respirability adequate drug payload’ There is something wrong with sentence.  I suggest it is rewritten again.

Reviewer 3 Report

This revised manuscript has been improved and the authors have addressed my concerns in the response. As a result, I agree that the manuscript may confer some useful information and is suitable for publication. There is one minor issue regarding the calculated aerodynamic diameters (using the method in section 2.2.9. Aerodynamic diameter). The calculated aerodynamic diameters were misleading and did not necessarily reflect the respirability. Such a calculation assumed the particles were fully dispersible upon aerosolization via a DPI and obviously it is unlikely to be true due to the facts that submicrosized particles are highly cohesive.
